# CRISPR/Cas9-Mediated Gene Editing of *BnFAD2* and *BnFAE1* Modifies Fatty Acid Profiles in *Brassica napus*

**DOI:** 10.3390/genes13101681

**Published:** 2022-09-20

**Authors:** Jianghua Shi, Xiyuan Ni, Jixiang Huang, Ying Fu, Tanliu Wang, Huasheng Yu, Yaofeng Zhang

**Affiliations:** Key Laboratory of Digital Upland Crops of Zhejiang Province, Institute of Crop and Nuclear Technology Utilization, Zhejiang Academy of Agricultural Sciences, Hangzhou 310021, China

**Keywords:** *BnFAD2*, *BnFAE1*, oleic acid, CRISPR/Cas9

## Abstract

Fatty acid (FA) composition determines the quality of oil from oilseed crops, and thus is a major target for genetic improvement. *FAD2* (*Fatty acid dehydrogenase 2*) and *FAE1* (*fatty acid elongase 1*) are critical FA synthetic genes, and have been the focus of genetic manipulation to alter fatty acid composition in oilseed plants. In this study, to improve the nutritional quality of rapeseed cultivar CY2 (about 50% oil content; of which 40% erucic acid), we generated novel knockout plants by CRISPR/Cas9 mediated genome editing of *BnFAD2* and *BnFAE1* genes. Two guide RNAs were designed to target one copy of the *BnFAD2* gene and two copies of the *BnFAE1* gene, respectively. A number of lines with mutations at three target sites of *BnFAD2* and *BnFAE1* genes were identified by sequence analysis. Three of these lines showed mutations in all three target sites of the *BnFAD2* and *BnFAE1* genes. Fatty acid composition analysis of seeds revealed that mutations at all three sites resulted in significantly increased oleic acid (70–80%) content compared with that of CY2 (20%), greatly reduced erucic acid levels and slightly decreased polyunsaturated fatty acids content. Our results confirmed that the CRISPR/Cas9 system is an effective tool for improving this important trait.

## 1. Introduction

Rapeseed (*B. napus*) is one of the most important oil-producing crops. The global production of rapeseed oil was 27.98 million tons during 2019 to 2020, making it the third largest source of vegetable oil (https://www.ers.usda.gov/data-products/oil-crops-yearbook/oil-crops-yearbook/ (accessed on 25 March 2022). Rapeseed oil is an important edible oil with high nutritional value [1], as well as a source for biodiesel fuels and raw materials in industry [2,3]. As an important breeding goal, the nutritional quality of oilseed oil is mainly determined by its fatty acid balance [4].

In rapeseed oil, the fatty acid profile contains palmitic (C16:0), stearic (C18:0), oleic (C18:1), linoleic (C18:2), linolenic (C18:3), eicosenoic (C20:1) and erucic (C22:1) acids [5]. In the seeds of the current “double low” (low erucic acid and low glucosinolate) rapeseed cultivar, the relative content of oleic acid is about 60%, and oil content ranges from 38–50%. Higher erucic acid content in rapeseed oil can cause myocardial damage and muscle lesions in the heart [6,7]. The higher oleic acid rapeseed oil not only increases oil stability at high temperature, prolonging shelf time [6,7], but also reduces cholesterol levels in rapeseed oil consumers, preventing the risk of arteriosclerosis and inflammatory diseases [8,9]. Therefore, genetic improvement should be carried out to promote oil quality, especially to further increase the oleic acid contents in rapeseed.

*FAD_2_, encoding fatty acid desaturase 2,* catalyzes the first committed step of the biosynthesis of polyunsaturated fatty acids from oleic acid to linoleic acid. *B. napus* (AACC, 2n = 38) is an allotetraploid species, containing four *FAD2* alleles, including *BnFAD2.A5*, *BnFAD2.A1* (a nonfunctional enzyme), *BnFAD2.C5* and *BnFAD2.C1.* [10]. Loss of function (4-bp insertion or single-nucleotide mutations in the coding regions) of *BnFAD2* genes [10,11,12], or mutations in *BnFAD2* genes generated by ethyl methanesulfonate treatment [13,14], resulted in increased oleic acid levels. The expression of *BnFAD2* can be depressed to increase the proportion of oleic acid in rapeseed using RNAi knockdown [15,16]. *FAE1*, encoding fatty acid elongase1, catalyzes the initial condensation step in the elongation pathway of very long chain fatty acid biosynthesis, and is thus a key gene in erucic acid bioxynthesis. There are two functional alleles of *FAE1* on the A8 and C3 chromosomes in *B. napus*. Expression of *BnFAE1* genes also closely relates to oleic acid content in rapeseed. Previous studies have reported that knockout of *FAE1* and knockdown of *FAE1* expression results in decreased erucic acid levels and increased oleic acid content [15,16,17,18]. Altogether, the studies discussed here show that *FAD2* and *FAE1* are suitable target genes for genetic manipulation to increase oleic acid content in rapeseed.

The CRISPR (Clustered regularly interspaced short palindromic repeats)/Cas9 system is an innovative targeted genome editing system [19,20,21], widely used in basic science and crop genetic improvement due to its simplicity, high efficiency, precision and low cost [22]. Unlike RNA interference, CRISPR/Cas9 technology does not necessitate the persistence of T-DNA, which, in most plant species, can be easily removed from the edited plants in successive generations by self-pollinating or backcrossing with receptor materials [23,24]. Because of these benefits, CRISPR/Cas9 technology has strong application prospects in crop breeding. There are a number of reports of successful genetic improvement and gene editing mediated by CRISPR/Cas9 in *B. napus*. For example, herbicide-resistant oilseeds rape has been created by editing of *BnALS1* gene [25], and rapeseed with multilocular siliques were obtained by editing of *BnCLV* [26]. Recently, novel mutants with high oleic acid content have been developed from double low (low erucic acid and low glucosinolate) rapeseed by editing of *BnFAD2* genes [27,28]. Double high (high oil and high erucic acid) rapeseed is also an important germplasm resource, but has not been extensively utilized to create new high oleic acid germplasm using CRISPR/Cas9 mediated gene editing.

In this study, we used the CRISPR/Cas9 system to create novel high oleic acid germplasms from the CY2 cultivar (which has almost 50% oil content of which approximately 40% is erucic acid). A CRISPR/Cas9 vector with two sgRNA expression cassettes was constructed to target *BnFAD2* and *BnFAE1* genes and agrobacterium-mediated genetic transformation was used to generate transgenic plants. The targeted mutations at *BnFAD2* and *BnFAE1* genes significantly increased oleic acid contents in the edited lines as compared with CY2. Our results showed that the CRISPR/Cas9 system is an effective targeted genome editing tool for genetic improvement, which can create desired novel germplasms used for rapeseed breeding.

## 2. Materials and Methods

### 2.1. Plant Materials and Growth Conditions

A semi-winter *B. napus* cultivar CY2 was used for Agrobacterium-mediated transformation [16]. Plants of CY2 and all related transgenic plants were grown under field conditions. In Haining, Zhejiang province, seeds were usually sown in late September or early October, and harvested in late May. The field experiment followed a randomized complete block with three replications. Each line was planted in one row with 10–12 plants, with a distance of 20 cm between plants within each row and 35 cm between rows. The field management was performed in line with standard breeding practice.

### 2.2. Plasmid Construction

We used the binary vector system pCAMBIA-1300 with a hygromycin selection marker to target the *BnFAD2* and *BnFAE1* genes. The target sequences used to generate sgRNA expression cassettes were designed using CRISPR-GE (http://skl.scau.edu.cn/targetdesign/ (accessed on 3 March 2020)) [29]. One sgRNA was able to specifically target *BnFAD2* (BnaC05g40970D), and another sgRNA could target two copies of *BnFAE1* (BnaA08g11130D and BnaC03g65980D) (Figure 1). The primers used to construct sgRNA in this study are listed in Appendix A.

The construction of CRISPR/Cas9 vectors including Cas9 and multiple sgRNA expression cassettes was performed according to the procedure described previously [19]. Briefly, sgRNA sequences were introduced into the sgRNA expression cassettes by overlapping PCR. Then the purified PCR products were integrated into pLYCRISPR/Cas9P35S-H via a Golden Gate construct.

To evaluate off-target potential in this study, the putative off-target sites were predicted using the CRISPR-P version 2.0 program [29].

### 2.3. Genetic Transformation of B. napus

The CRISPR/Cas9 vector was transformed into Agrobacterium tumefaciens strain GV3101 through the freezing and thawing method. Then the editing vector was transformed into CY2 plants by hypocotyl segment transformation as described previously [30,31].

### 2.4. Mutation Analysis

Genomic DNA was extracted from selected plants using the hexadecyl trimethyl ammonium bromide (CTAB) method [32]. T0 plants were obtained by hygromycin (50 mg/L), and the presence of T-DNA elements was investigated by PCR with Cas9 specific primers [27]. The transgenic plants were grown in an isolated nursery at 25 °C under long-day conditions (16 h light/8 h night). The primers used to amplify the target regions were designed in the flanking region of the Cas9/sgRNA targets. The targeted mutagenesis was analyzed in the edited lines (T_0_ to T_4_ generation) by PCR amplification and Sanger sequencing. PCR conditions were as follows: For *BnFAD2*, 95 °C for 2 min, followed by 35 cycles at 94 °C for 30 s, 53 °C for 30 s, 72 °C for 1 min and a 7 min final extention at 72 °C; For *BnFAE1* (BnaA08g11130D), 95 °C for 2 min, followed by 35 cycles at 94 °C for 30 s, 54 °C for 30 s, 72 °C for 1 min and a 7 min final extension at 72 °C. For *BnFAE1* (BnaC03g65980D), 95 °C for 2 min, followed by 35 cycles at 94 °C for 30 s, 55 °C for 30 s, 72 °C for 1 min and a 7 min final extension at 72 °C. The primers used for PCR amplification are listed in Appendix A.

### 2.5. Analysis of Lipids

For analyzing lipid levels, seeds were collected from plants (CY2 and the edited plants) grown under the same conditions and then stored in a freezer until they were analyzed. Analysis of the fatty acid levels was performed by GC–MS as described in [17]. Relative fatty acid compositions were calculated as the percent of each fatty acid relative to total measured fatty acids. The seed oil content was analyzed by the Soxhlet extraction method as described previously [33].

## 3. Results

### 3.1. CRISPR/Cas9-Mediated Editing of BnFAD2 and BnFAE1

To create high oleic acid oilseed rape, targeted mutations were generated to the *BnFAD2* and *BnFAE1* genes in CY2 cultivar. Using CRISP-GE (http://skl.scau.edu.cn/home/ (accessed on 1 May 2020)) [34], one single-guide RNA (sgRNA) was designed to target *BnFAD2* (BnaC05g40970D), and the other was designed to simultaneously target two copies of *BnFAE1* (BnaA08g11130D and BnaC03g65980D) (Figure 1a). A single editing vector (Figure 1b) that targeted *BnFAD2* and *BnFAE1* with two sgRNA expression cassettes was constructed using a CRISPR/Cas9 basic vector as described previously [19]. The CRISPR/Cas9 vector used in this study contains a hygromycin phosphotransferase (HPT) selection marker, sgRNA transcriptions and a Cas9 nickase. sgRNAs were driven by the Arabidopsis U3d and U3b promoter, respectively.

The editing vector was transformed into *B. napus* ‘CY2′ using the Agrobacterium tumefaciens-mediated hypocotyl method, generating 30 independent T_0_ plants. All independent T_0_ plants were positive transformants harboring T-DNA insertions from PCR analysis using Cas9-specific primers [27]. The target regions of *BnFAD2* and *BnFAE1* were amplified by PCR using gene specific primers and were subjected to Sanger sequencing. Based on the sequencing results, twelve, six and five out of 30 T_0_ plants exhibited editing events in the target sites of BnaC05g40970D (*BnFAD2*), BnaA08g11130D (*BnFAE1*) and BnaC03g65980D (*BnFAE1*), respectively (Table 1). Three lines (Line 6, Line11 and Line 14) showed mutations in all three target sites of BnaC05g40970D, BnaA08g11130D and BnaC03g65980D (Table 1).

To evaluate off-target potential in this study, the putative off-target sites were predicted using the CRISPR-P version 2.0 program [35]. Then 15 putative off-target sites for both *BnFAD2* and *BnFAE1* were identified by the program with default settings. Of these, seven putative off-target sites located at the CDS coding sequences were chosen for further analysis. The sequencing results for 16 edited plants did not show any editing events (Table 2).

### 3.2. Mutations at BnFAD2 and BnFAE1 Altered Fatty Acid Profiles in the Edited Plants

In the edited plants listed in Table 1, three lines (Line 6, Line 11 and Line 14) exhibited editing events in three target sites of BnaC05g40970D, BnaA08g11130D and BnaC03g65980D. The three lines were followed for the next four generations. Homozygous three-target mutation lines (Line 6-9-7-3-1, Line 14-7-12-4-9 and Line 11-9-18-7-4) were identified (Figure 2). The mutation sites remained the same at the target sites of BnaC05g40970D, BnaA08g11130D and BnaC03g65980D through generations (Figure 2).

To characterize the effect of editing events of *BnFAE1* and *BnFAD2* on seed fatty acid levels, mature seeds from the edited plants (Line 6-9-7-3-1, Line 14-7-12-4-9 and Line 11-9-18-7-4) were subjected to fatty acid analysis (Table 3). The level of oleic acid of CY2 seeds was about 21.4% under field conditions. The oleic acid contents of the three edited plants were significantly increased in their seeds, ranging from 73.3% to 76.47% (Table 3). The level of erucic acid in the edited plants was greatly decreased compared with that in CY2 (Table 3). Levels of linoleic acid, linolenic acid and eicosenoic acid were decreased to some extent in the edited plants (Table 3).

### 3.3. Oil Content in Seeds of the Edited Plants

To examine the effects of gene editing of *BnFAE1* and *BnFAD2* on seeds’ oil content, we measured the seed oil content in the edited plants. The oil content from CY2 seeds was about 50.28% dry weight (Figure 3 ). The oil contents in the seeds of the edited plants were between 43.56 and 47.77%, decreased by 5–13.4% relative to CY2 seeds (Figure 3).

## 4. Discussion

The CRISPR/Cas9 system is a convenient and highly efficient multiplex genome editing system, which has been utilized in a variety of species including *B. napus* [19,24,36]. In this study, we designed two sgRNA to target *BnFAD2* and *BnFAE1* genes in the CY2 cultivar (Figure 1). The results of gene editing showed that induced mutations contained insertions, deletions and substitutions at the target region in *BnFAD2* and *BnFAE1* genes (Table 1). Moreover, mutations in *BnFAD2* and *BnFAE1* genes significantly altered fatty acid profiles in the edited lines, resulting in increased oleic acid levels and decreased erucic acid content (Table 3). Taken together, our results show that induced mutations by CRISPR/Cas9 mediated gene editing in *BnFAD2* and *BnFAE1* genes greatly improve the nutritional quality of the oil from CY2 cultivar.

Concerning the CRISPR/Cas9 technique, editing efficiency is an important evaluation index, which is affected by plant codon optimization of Cas9, the expression levels of Cas9 and sgRNA, the sequence composition (such as GC content) of targets and the secondary structure of the target-sgRNA [19]. In this study, two sgRNAs were designed to target the *BnFAD2* and *BnFAE1* genes. The editing efficiency at the targets of the *BnFAD2* gene and two copies of *BnFAE1* gene ranged from 16.7–40% (Table 1). A previous study, examining the mutation efficiency of 12 genes by the CRISPR/Cas9 method, showed that the mutation efficiency at each target site for sgRNA-mediated mutagenesis ranged from 5.3% to 96.6% [37]. Considering the major influence of Cas9 and sgRNA on the editing efficiency [38,39], optimizing plant promoters for Cas9 and effective targets for sgRNA (such as higher GC contents and fewer secondary structures) may significantly increase the frequency of mutagenesis.

A number of studies have been performed to increase oleic acid levels in rapeseed oil using different techniques. RNAi knockdown of *BnFAD2* and *BnFAE1* resulted in increased oleic acid (80%) and decreased polyunsaturated fatty acids (PUFA; 9%) in *B. napus* [15]. Knockout of the *FAD2-1A* and *FAD2-1B* genes by TALEN technique resulted in increased oleic acid and decreased PUFA in soybean [40]. Recently, CRISPR/Cas9 mediated gene editing of *BnFAD2* alleles also altered the fatty acid profile in *B. napus*, resulting in increased oleic acid [27,28]. Although novel mutants with higher oleic acid content have been created by different techniques, a few unfavorable agronomic traits were found in them. When the *FAD2* gene in Arabidopsis was knocked out, the plants could not grow well at low temperature (6–12 °C) [41]. Double mutants of *fad2fae1* in Arabidopsis showed a stunted bushy phenotype and reduced seed production [42]. In *Camelina sativa*, a triple mutant of the *FAD2* gene showed a stunted bushy phenotype and very low seed yield [43], suggesting that complete inactivation of the *FAD2* genes might affect plant development. In *B. napus*, mutants with more than 80% oleate in the seed oil had significantly lower seedling establishment and vigor, delayed flowering and reduced plant height at maturity under field conditions [13]. Furthermore, these mutants showed 7–11% reductions in seed oil content. In this present study, we generated mutants with increased oleic acid (70–80%) and decreased PUFA (13–16%) by editing one copy of *BnFAD2* gene and two copies of *BnFAE1* gene, which displayed normal phenotype in the field. All these studies indicate that genetic improvement to further increase oleic acid content should maintain suitable PUFA content in fatty acid profiles, and more research should be done to examine the minimal content of the PUFA required for plant growth and development in most plant species.

High oil content and high oleic acid levels are both significant agronomic traits, being the important goals of rapeseed breeding. In the present study, levels of oleic acid were greatly increased due to CRISPR/Cas9 mediated editing of *BnFAD2* and *BnFAE1* genes; however, the oil content was decreased to some degree in the edited plants (Figure 3). The phenomenon that gradually increased oleic acid content is associated with the decrease of the oil content to some degree in one specific genetic material have been reported in previous studies [13,16]. To develop the “double high” (the high oil and high oleic acid) rapeseed cultivar, novel genetic factors should be identified and integrated into conventional cultivars by crossing or genetic manipulation.

## 5. Conclusions

In summary, our results demonstrated that induced mutations by CRISPR/Cas9 mediated editing of *BnFAD2* and *BnFAE1* genes could generate novel high oleic acid germplasm from CY2 cultivar (high oil and high erucic acid), and that the CRISPR/Cas9 system is an effective tool for basic research and crop genetic improvement in B. napus.

## Figures and Tables

**Figure 1 genes-13-01681-f001:**
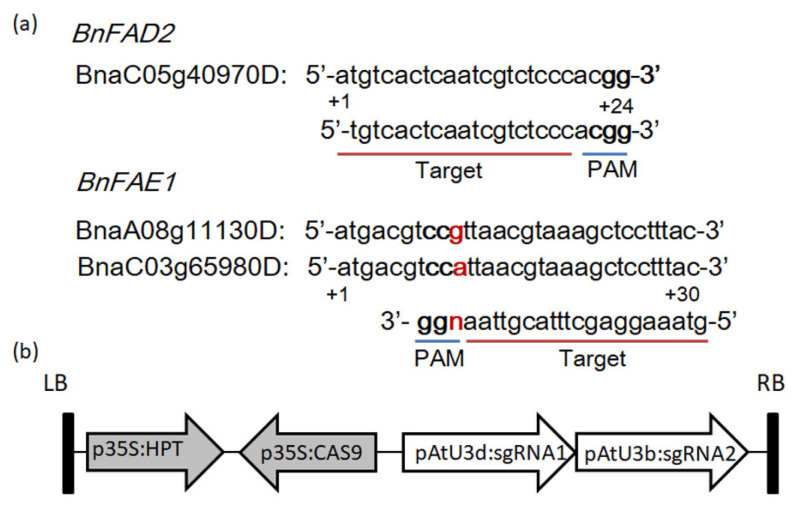
Target sequences of *BnFAD2* and *BnFAE1* genes and vector construction. (**a**) Sequences of the target sites in *BnFAD2* and *BnFAE1* genes are shown. PAM sequences (NGG) are indicated in bold font. (**b**) Schematic of the genome editing vector. The vector contains a hygromycin phosphotransferase (HPT) gene as a selection marker and the 35S promoter (p35S) to drive CAS9.

**Figure 2 genes-13-01681-f002:**
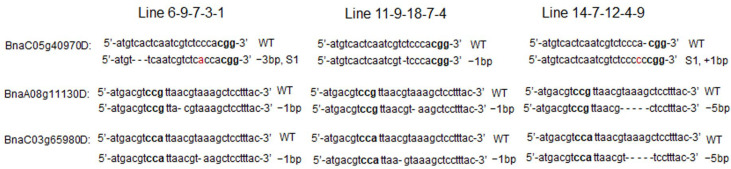
Mutations in the target region of *BnFAD2* and *BnFAE1* in homozygous mutants. Substitutions, insertions and deletions are indicated in red fonts, “+” and “−”, respectively. S1 indicates a nucleotide substitution. PAM was highlighted in bold fonts.

**Figure 3 genes-13-01681-f003:**
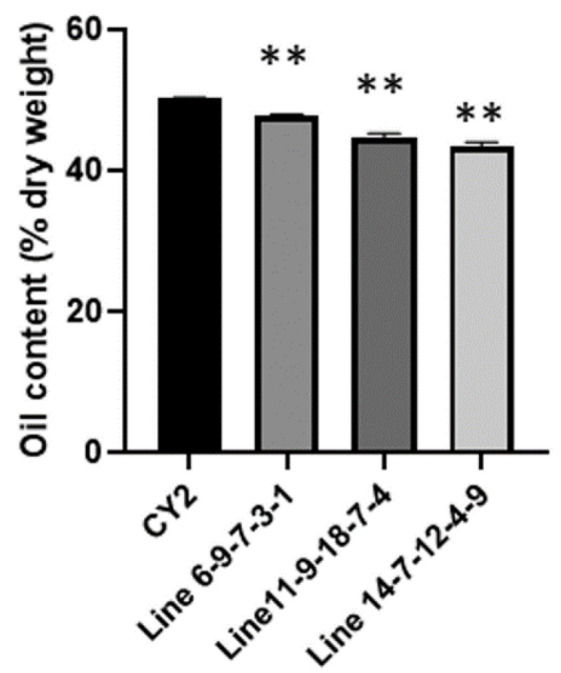
Oil content in seeds of the T_4_ generation edited lines. Mature seeds were used for analysis of the oil content by the Soxhlet method. Values are means ± SE (*n* = 3). Asterisks indicate statistically significant differences compared with the control (Student’s *t* test: *** p* < 0.01).

**Table 1 genes-13-01681-t001:** Genotype of *BnFAD2* and *BnFAE1* genes in T_0_ edited plants.

Line ID	*BnFAD2*	*BnFAE1*	
	BnaC05g40970D	BnaA08g11130D	BnaC03g65980D
Line 1	+1 bp	WT	WT
Line 5	WT	−1 bp	WT
Line 6	−3 bp, S1	−1 bp	−1 bp
Line 7	+1 bp	WT	WT
Line 8	−2 bp	WT	WT
Line 11	−1 bp	−1 bp	−1 bp
Line 12	S1	WT	WT
Line 14	S1, +1bp	−5 bp	−5 bp
Line 15	−2 bp	WT	WT
Line 17	WT	WT	−1 bp
Line 19	−1 bp	WT	WT
Line 22	WT	−2 bp	WT
Line 24	WT	WT	−1 bp
Line 25	−1bp	WT	WT
Line 27	−1 bp	+1 bp	WT
Line 29	+1 bp	WT	WT

“+’ and “−” represent the insertion and deletion of the stated number of nucleotides, respectively. S1 indicates a nucleotide substitution.

**Table 2 genes-13-01681-t002:** Detection of putative off-target potential at each sgRNA target in T_0_ edited plants.

Gene (Locus)	Sequence	Putative Off-Target Location	No. of Plants Sequenced	Off-Target Editing
on-target				
*FAD2* (BnaC05g40970D)	TGTCACTCAATCGTCTCCCA **CGG**			
off-target				
1	TGTCACTCAATCGTCTC**A**CA **CGG**	BnaA05g26900D	16	no
2	TGTC**G**CT**TCC**TCGTCTCCCA **CGG**	BnaC09g54430D	16	no
3	TGTCAC**C**CAAT**G**GTCTCCC**T CGC**	BnaC04g05410D	16	no
4	T**T**T**A**ACTCAATC**TC**CTCCCA **CGA**	BnaA05g28940D	16	no
on-target				
*FAE1* (BnaA08g11130D)	GTAAAGGAGCTTTACGTTAA **CGG**			
off-target				
1	**T**T**C**AA**C**GAGCTT**G**ACGTTAA **CGG**	BnaA07g33300D	16	no
on-target				
*FAE1* (BnaC03g65980D)	GTAAAGGAGCTTTACGTTAA **TGG**			
off-target				
1	GT**G**AAGGA**T**CTTTACGT**A**A**C** **TGG**	BnaCnng35470D	16	no
2	GTAAAGGA**AG**TT**G**A**A**GTTAA **TGG**	BnaA04g10800D	16	no

The red base in the off-target sequence represents mismatch to on-target, and PAM sequences are indicated in bold font. The green base in the off-target sequence represents match to PAM.

**Table 3 genes-13-01681-t003:** Relative FA content in seeds of the edited lines.

Lines	Generation	Relative FA Content (%)						
		C16:0	C18:0	C18:1	C18:2	C18:3	C20:1	C22:1
CY2	T4	3.09 ± 0.13	1.07 ± 0.12	21.4 ± 0.48	12.26 ± 0.44	7.13 ± 0.37	11.53 ± 0.58	42.58 ± 0.53
Line 6-9-7-3-1	T4	3.67 ± 0.06 *	ND	73.3 ± 0.34 **	9.43 ± 0.23 **	6.8 ± 0.35 *	3.13 ± 0.2 **	2.35 ± 0.18 **
Line 11-9-18-7-4	T4	3.5 ± 0.06 *	ND	75.43 ± 0.39 **	8.37 ± 0.2 **	6.67 ± 0.24 *	3.07 ± 0.07 **	2.1 ± 0.17 **
Line 14-7-12-4-9	T4	3.37 ± 0.09	ND	76.47 ± 0.35 **	7.43 ± 0.32 **	6.23 ± 0.15 *	3.07 ± 0.18 **	2.04 ± 0.1 **

Relative FA contents were calculated as the percentage that each fatty acid represented of the total fatty acid profile. Mature seeds were used for the analysis. Values are means ± SE (*n* = 3). Asterisks indicate statistically significant differences compared with the control (Student’s *t* test: ** p* < 0.05; *** p* < 0.01). ND, Not detectable.

## Data Availability

The original contributions presented in the study are included in the article/Appendix A, further inquiries can be directed to the corresponding authors.

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
