# Peer review of "CRISPR/Cas9-Mediated Gene Editing of BnFAD2 and BnFAE1 Modifies Fatty Acid Profiles in Brassica napus"

_genes, 2022, doi:10.3390/genes13101681_

Round 1

Reviewer 1 Report

In the file attached it can be found some minor suggestions for improving the MS. Quality of tables and captions should be improved. 

Reviewer 2 Report

The present paper is concerned with the study of CRISPR/Cas9 method as a convenient tool for genetic improvement of plants, including as a tool to expand the available germoplasm, which can be used to develop breeding programmes using classical methods. The main focus of this work is on trying to use genetic editing of genes involved in the control of fatty acid biosynthesis in order to potentially improve rapeseed oil as an edible oil. Genetic modifications of these genes have been attempted previously and have been successful. The novelty of this work is the use of genetic editing to simultaneously alter the two genes FAD2 and FAE1. The authors have clearly demonstrated the feasibility and effectiveness of this method for potential genetic improvement of rapeseed oil quality. Namely, it was possible to considerably increase oleic acid content and decrease erucic acid content compared to the original cultivar grown under the same conditions. The discussion sections provide some insight into the possible further applications and potential improvement of this technique rapeseed.

Introduction - The introduction is written clearly, concisely and is well-structured. All provided information is relevant to the study. However, in my opinion, it still lacks some information that is relevant and might help to point out more clearly the significance of the research. In particular, Introduction would benefit from adding the following information:

1) relative content of fatty acids in rapeseed oil and total oil content range;

2) since, two genes controlling two different biosynthesis pathways in fatty acid metabolism are studied in the work, it would be worth providing information on how the major fatty acids, not only oleic acid, impact the nutritional qualities of rapeseed oil;

3) brief description of FAD2 enzyme and its role in fatty acid biosynthesis, and therefore, in the control of relative C18 fatty acid content; brief description of the four copies of the FAD2 gene in B. napus, including their genome localisation, functionality/nonfunctinality etc.

4) adding the same information about the BnFAE gene (no of copies, localisation, functionality, sequence similarity, etc.) and the protein it encodes (what role it plays in fatty acid metabolism) would be also highly recommended. It would be useful if the information on how FAE enzyme may have an impact on oleic acid content could be added as well.

Line 36. I would suggest to rewrite the phrase "but also reduces cholesterol levels, preventing the risk of arteriosclerosis inflammatory diseases [8, 9]" so that it were clear that "cholesterol levels" are reduced not in the plants, but in rapeseed oil consumers. currently, it sounds a bit confusing. Also, "and" should be inserted between "arteriosclerosis" and "inflammatory diseases" (line 37).

Lines 41-42. I would recommend again to rewrite the sentence in such a way that it would be clear that mutations in the first half of the sentence are naturally occurring ones since it might  be confusion for the reader. In the view, of what was mentioned above it may be useful to indicate what kind of mutations are these, I mean loss of function, or gain of function?

Line 52. I would recommend specifying "low cost", or "cost efficiency" here.

Lines 53-55. It might be better to change the phrase "persistent existence" to "persistence".

Materials and Methods - The section is generally well-written, it contains mostly all the information necessary to understand how the work was carried out in the technical terms. However, I would suggest to expand certain subsections.

Subsection 2.1. Could you, please, provide more information on how the plants were grown in the field.

Subsection 2.2. I would recommend to transfer the technical data on checking for putative off-target site from the Results section here leaving only the actual results in the Results section. What is the source of nucleotide sequences that were used to design sRNAs? Have you sequenced the gene, or used the publicly available data?

It looks like the reference [29] may not be accurate. Please, check it, as it refers to the paper Ma, X.; Chen, L.; Zhu, Q.; Chen, Y.; Liu, Y.G. Rapid Decoding of Sequence-Specific Nuclease-Induced Heterozygous and Bial-321 lelic Mutations by Direct Sequencing of PCR Products. Mol Plant. 2015, 8,1285-1287 which discusses the post-sequencing procedures to differentiate between the resulting homo-/heterozygous plants, and not to gRNA design itself.  If you used this method for post-sequencing differentiation, it might be useful to add this link to the following section. In the Materials and Methods, it could be recommended to provide a reference to the original paper (Xie X, Ma X, Zhu Q, Zeng D, Li G, Liu Y-G. 2017. CRISPR-GE: A Convenient Software Toolkit for CRISPR-Based Genome Editing. Mol. Plant 10(9):1246-1249).

Caption to Figure 1 seems to be duplicated.

Subsection 2.4. I would recommend to expand this subsection a little. What was the concentration of antibiotic in the selection medium? Could you, please, specify the growth conditions of regenerated plants? Could you, please, specify PCR amplification conditions for mutation events verification.

Primers used to check the presence of Cas9 in the transformed plants are not listed in the Supplementary Table S1.

Line 114. It looks like the preposition "in" is lacking.

Results - The results are clearly presented and conclusive. The only concern which may be expressed is using just three plants to measure the relative values of fatty acid and total oil content.

Subsection 3.1. Could you, please, provide justification for choosing these particular copies of FAD2 and FAE1 genes for gene editing in the present work. Also, it would be useful if some details could be provided on the design of sgRNA targeting the studied genes (in particular, localisation of target sites). Did you verify that only the target copies of the BnFAD2 and BnFAE1 gamily were mutated?

It is unclear why initially there were 40 T0 plants and then further in the paper they become 30 plants notwithstanding the fact that all 40 plants  proved to be positive for T-DNA insertion. Could you, please, check it carefully.

Did you check for the status of mutations in the resulting T0 plants and further generations? If it has been done, could you, please, provide the details in the corresponding sites of Materials and Methods and Results sections.

I would recommend to combine the Results subsections 3.1 and 3.2 into a single subsection transferring the technical detains on off-target sites check to the Materials and Methods section. 

It would be interesting to know the consequences of the mutations introduced in the FAD2 and FAE1 gene in terms of translated proteins. Did you check whether they were stop-codons, or frame shift, or a nonsynonymous substitution?

Subsection 3.3. Could you, please, specify how the three gene-edited lines containing mutations in all three target genes were maintained for four generations? Was verification of the stability of the introduced mutations performed in every generation? Could you specify in which way it was done?

Did you check fatty acid and seed oil content only in the three selected lines only in the fourth generation? Have you checked the relative content of different fatty acids and general oil content in the seeds of T0 plants?

Caption to Figure 2 seems to be duplicated.

Subsection 3.4. I may recommend to indicate that total oil content was measured in the gene-edited plants of 4th generation since it is somewhat confusing as it reads now.

Caption to Figure 3 seems to be duplicated.

Discussion - The section is well-presented. Interesting points have been risen which could possibly lead to further improvements of the technique to successfully use it for edible rapeseed oil quality improvement. Regarding the discussion on possible means to increase the efficiency of gene editing, I may recommend also trying other promoters for sgRNA production.

A discussion of the secondary effects that mutations in the FAD2 and FAE1 genes have on the physiological processes in plants and on related phenotypic traits is very interesting and useful. In this connection, I wonder whether you compared the relative fatty acid content and total oil content  and checked the overall phenotypes in the gene-edited plants containing only one mutation in the FAE1 gene, for example in lines 17, 24, 27? Also, do you not link the general reduction in oil content to the effect of mutations in the FAE1 gene, namely with the reduction in erucic acid content?

Lines 201-204, I would recommend rephrasing to covey the idea more clearly.

Line 213. It looks like it should read "rapeseed oil", not "oilseed oil".

Line 221. The phrase "the only FAD2 gene" is not clear. Could you, please, rewrite it so that the meaning was more transparent. 
